# The Mediating Role of Dispositional Optimism in the Relationship between Health Locus of Control and Self-Efficacy in Pregnant Women at Risk of Preterm Delivery

**DOI:** 10.3390/ijerph19106075

**Published:** 2022-05-17

**Authors:** Iwona Niewiadomska, Agnieszka Bień, Ewa Rzońca, Krzysztof Jurek

**Affiliations:** 1Department of Social Psychoprevention, John Paul II Catholic University of Lublin, 20-950 Lublin, Poland; iwona.niewiadomska@kul.pl; 2Chair of Obstetrics Development, Faculty of Health Sciences, Medical University of Lublin, 4/6 Staszica St., 20-081 Lublin, Poland; agnieszka.bien@umlub.pl; 3Department of Obstetrics and Gynecology Didactics, Faculty of Health Sciences, Medical University of Warsaw, 14/16 Litewska St., 00-575 Warsaw, Poland; erzonca@wum.edu.pl; 4Institute of Sociological Sciences, John Paul II Catholic University of Lublin, 20-950 Lublin, Poland

**Keywords:** pregnancy, threatened preterm labor, dispositional optimism, self-efficacy, health control, personal resources, conservation of resources (COR) theory

## Abstract

Difficult situations during pregnancy, such as threatened preterm labor, trigger negative experiences in women. The levels of stress experienced and the way individuals cope with it depend on their personal resources, such as optimism, internal health locus of control, and self-efficacy, among other factors. The purpose of this paper was to determine the role of dispositional optimism in the relationship between health locus of control and self-efficacy in pregnant women with threatened preterm labor. Dispositional optimism plays the role of mediator in relationships between: (1) internal health control and self-efficacy; and (2) impact of random events on one’s health and self-efficacy. Dispositional optimism does not mediate the relationship between the perceived impact of others on one’s health and self-efficacy. For women with a high-risk pregnancy, dispositional optimism is a significant resource for coping with the problems they encounter. It changes the direction (from negative to positive) of the association between experiencing the impact of external factors (random events) on one’s health and perceived self-efficacy. It prevents the cycle of loss caused by the interpretation of random events as having an impact on one’s health, and acts as a mediator to initiate a cycle of gains that leads to greater perceived self-efficacy. Optimistic pregnant women maintain a positive outlook, even when confronted with difficult, negative experiences such as threatened preterm labor.

## 1. Introduction

Pregnancy is an exceptional time in a woman’s life. During that time, she undergoes a number of changes, both physical and psychological, and may experience a range of emotions influenced by hormonal, anatomical, and adaptive processes. Though the pregnancy period is commonly viewed as a joyful time of expectation, it may become difficult, especially if there is a risk of preterm labor [1,2]. Such situations may result in more negative experiences, changing the way the pregnancy is perceived, leading to feelings of loss of control, fear for oneself and one’s baby, and high levels of stress. The stress experienced and the way individuals cope with it depend, among other factors, on one’s personal resources [2]. These include optimism, internal health locus of control, and self-efficacy.

Dispositional optimism is defined as a relatively stable belief that human life is generally associated with positive outcomes, and that the negative aspects of the present have the potential for positive evolution in the future. This belief comprises, on the one hand, perceiving, explaining, and assessing the world and the phenomena observed in positive rather than negative terms, and on the other, anticipating favorable rather than adverse future events [3,4,5,6].

Research findings suggest that high levels of dispositional optimism are significantly associated with physical and mental health, life expectancy, high motivation in one’s tasks (even when faced with growing obstacles), positive emotions, more self-efficacy, higher perceived quality of life, resistance to stress, preference for active coping strategies, and a less severe negative impact of stress [3,7,8,9,10,11,12].

Locus of control is seen as a personality dimension responsible for one’s generalized way of defining causal relationships between one’s behaviors and their consequences. An internal locus of control implies the belief that positive and negative life events are the consequence of one’s own actions and are subject to one’s personal control. Conversely, an external locus of control means that positive or negative life events are seen as unrelated to one’s own behavior and thus considered beyond one’s control. Those with an external locus of control are convinced that their actions are of little consequence, since their fate is determined by external factors such as randomness, luck, other people, or institutions [13,14,15,16].

Generalized self-efficacy is the belief in one’s own capability to mobilize one’s cognitive resources, competences, learned solutions, and efforts required to effectively perform tasks. Individuals with a stronger sense of self-efficacy are more likely to see the obstacles encountered and/or goals that have been set as a sort of a challenge to be taken up. Self-efficacy is considered a self-regulation skill involved in setting ambitious and important goals, as well as in decision making, motivation to act, and involvement in one’s actions. The regulation function of self-efficacy relies on the association between high levels of this variable and perceived success/satisfaction with one’s performance, constructive coping with difficulties, and increased perceived quality of life. In turn, low levels of self-efficacy considerably reduce motivation potential and commonly co-occur with depression, anxiety, and helplessness [12,17,18,19].

From the perspective of S. Hobfoll’s conservation of resources (COR) theory, these variables, locus of control and self-efficacy, are a category of personal resources that regulate human behavior in a way dependent on a number of mechanisms. One of these mechanisms is described by the first COR principle, whereby resource loss affects a person’s functioning disproportionately more than resource gain. Thus, an individual will undertake various actions to prevent the loss of the adaptive resources they have, among other examples, by initiating gain cycles to compensate for any losses. Another mechanism is that in a situation of perceived loss, individuals tend to focus on any gains they have achieved and/or could achieve to reduce or mitigate the negative effects of the stress they experience. One could therefore conclude that women hospitalized due to high-risk pregnancy experience personal resource losses that generate stress, which in turn poses a high risk of further resource losses (in what is referred to as a loss cycle). COR theory explains adopting a passive attitude towards the problems one encounters. According to S. Hobfoll, the situation is worst for those with scarce resource reserves, as their resource deficits prompt them to adopt a defensive attitude in the face of any difficulties. This is due to the fact that an individual with extremely limited resources experiences very intense stress, e.g., in the form of strong negative emotional reactions to initial resource losses. Each subsequent loss exacerbates this psychological tension and enhances negative emotions. As a consequence, the need to counteract unpleasant psychological states drains the existing adaptive reserves [20,21].

In accordance with the second COR principle, the stress one experiences may contribute to another important mechanism, whereby one invests one’s personal resources to trigger resource gains. In line with this principle, an individual initiates a process of resource acquisition to prevent resource loss, compensate for any loss that has already occurred, and/or gain new resources. One important factor involved in this cycle of gains in a stressful situation is the possession of reserves (an adequate level of resources), which, on the one hand, limit the negative consequences of any stress experienced, and on the other, stimulate activities leading to a cycle of resource gains, such as investing existing resources in behaviors allowing one to solve their problems and/or achieve important personal goals [12,20,21,22,23,24,25].

To explain the relationships between the psychological variables tested in our study, we used the COR concept of resource conservation, which is currently regarded as an important theory for understanding stress (both traumatic and chronic) and building psychological resilience [22,26]. In the processual explanation of stress and resilience, important functions are attributed to subjective resources such as self-efficacy, sense of control, and optimism, among others [24]. COR theory provides a strong theoretical framework for the study of stress, its consequences, and its relationship with the psychological capital, and is an expansive theory of human behavior based on the key need to acquire and conserve resources in order to survive. Dispositional optimism could be then considered the resource of resilience: it is related to positive mood, to perseverance and effective problem solving, to personal success, to good health status and to long life. In contrast, pessimism is related to depression, failure, social estrangement, morbidity, or mortality. The use of COR theory for the interpretation of the study results should lead to making them relevant to mechanisms developed from established theoretical approaches in the literature [22,26].

The observations described above warrant the formulation of the following research hypotheses regarding the relationships between the personal resources that were studied in women hospitalized due to high-risk pregnancy:

Dispositional optimism in women with high-risk pregnancy is a mediator in the relationship between health locus of control and perceived self-efficacy.
**Hypothesis** **1.** *Dispositional optimism mediates the relationship between the perceived personal impact on one’s health and self-efficacy.*
**Hypothesis** **2.** *Dispositional optimism mediates the relationship between the perceived impact of others on one’s health and self-efficacy.*
**Hypothesis** **3.** *Dispositional optimism mediates the relationship between the perceived impact of random events on one’s health and self-efficacy.*

To the best of our knowledge, the role of dispositional optimism as a mediator in the relationship between the locus of health control and the sense of self-efficacy among women in complicated pregnancy has not been analyzed, and our study is the first of its kind.

### Purpose of the Study

The purpose of this paper was to determine the role of dispositional optimism in the relationship between health locus of control and self-efficacy in pregnant women with threatened preterm labor.

## 2. Materials and Methods

### 2.1. Study Groups

The study was performed in the years 2017–2020 in a group of 328 pregnant women hospitalized due to threatened preterm labor. Inclusion criteria were: consent to participate in the study, age above 18 years (the legal age of majority in Poland), Caucasian race, speaking Polish as one’s native language, singleton pregnancy, and receiving proper prenatal care since the beginning of the pregnancy. Exclusion criteria included multiple pregnancy or diagnosis of a lethal fetal anomaly (Figure 1).

The most numerous group was women aged 26–35 years (57.9%), urban residents (54.6%), married/in a stable relationship (72.3%), had not completed higher education (51.2%), and had a satisfactory socioeconomic standing (51.8%); typically, they were pregnant for the second time (44.5%), nulliparous (76.5%), between weeks 32 and 37 (36.3%)—Table 1.

The study was approved by the Lublin Medical University Bioethics Committee (approval no. KE-0254/284/2017). Each participant was informed about the purpose of the study and provided with questionnaire completion instructions. Respondents were informed that participation was voluntary, and that the study results were anonymous and to be used exclusively for research purposes. All respondents provided their informed consent in writing.

### 2.2. Assessments

The study used a diagnostic survey with questionnaires. The following instruments were applied: the life orientation test (LOT-R), the multidimensional health locus of control scale (MHLC), the generalized self-efficacy scale (GSES), and a standardized interview questionnaire with items concerning the participants’ characteristics.

The Revised Life Orientation Test (LOT-R) by Scheier, Carver, and Bridges, was adapted for use in Polish settings by Poprawa and Juczyński. This test evaluates the respondent’s dispositional optimism based on 10 statements, six of which are diagnostic, while four are filler items. Each is rated on a five-item scale, from 0—strongly disagree, to 4—strongly agree. Total scores range between 0 and 24 points. Scores of 17–24 points indicate a high level of dispositional optimism, 13–16—moderate optimism, and 0–12—a pessimistic disposition. Cronbach’s alpha for the scale’s internal consistency is 0.76 [27].

The multidimensional health locus of control scale (MHLC) by Wallston, Wallston, and DeVellis, was adapted for use in Polish settings by Juczyński. It comprises 18 statements referring to generalized expectations in three health locus of control dimensions: internal (I am in control of my health), external factors/impact of others (my health results from the actions of others, including medical personnel), and the belief that one’s health depends on random events. Each item of each subscale is rated between 1 and 6 points. Points are totaled in each MHLC dimension. Scores are interpreted based on the median value: those above the median are considered high, and those below the median are considered low. Scale reliability is 0.64 for internal control, 0.59 for impact of others, and 0.63 for random events [28].

The generalized self-efficacy scale (GSES) by Schwartzer and Jerusalem, was adapted for use in Polish settings by Juczyński. The scale measures an individual’s general perception of their efficacy in dealing with difficult situations and obstacles. The GSES comprises 10 items rated between 1 and 4 points. Higher scores indicate a greater sense of self-efficacy. Scores between 10 and 24 points are interpreted as indicating a low level of self-efficacy, 25–29 points as moderate, and 30–40 as showing a high level of self-efficacy. Cronbach’s alpha for scale reliability is 0.85, and the internal consistency of the GSES ranges between 0.76 and 0.91 [29].

### 2.3. Statistical Analyses

We performed a series of correlation analyses and mediation analyses (three models). First, we analyzed zero-order correlations among the variables. The mediation analysis was performed in accordance with the guidelines provided by Preacher and Hayes, using IBM SPSS Statistics 26 (IBM Corp., Armonk, New York, NY, USA) and PROCESS macro for SPSS (written by Andrew F. Hayes).

We tested the significance of indirect effects using the bootstrapping procedure. Unstandardized indirect effects were computed for each of the 5000 bootstrapped samples. We used bias-corrected and accelerated 95% confidence intervals. If zero is not included in the 95% CI, the mediating role (c′) is statistically significant (Figure 2).

Figure 1 depicts a simple mediation model with one mediator. Path c represents the total effect of the sense of responsibility for one’s health (MHLC; three separate models: internal, external, and accidental) on the sense of perceived self-efficacy (GSES). Path c′ represents both the direct effect of the sense of responsibility for one’s health on the sense of perceived self-efficacy and the indirect effects of the sense of responsibility for one’s health on the sense of perceived self-efficacy via dispositional optimism (LOT-R) as a mediator. The specific indirect effect is the product of a and b (path a × path b).

We used the Monte Carlo Power Analysis for Indirect Effects application developed by Schoemann, Boulton, and Short. A post hoc Monte Carlo power analysis for the indirect effect (with 1000 replications and 20,000 draws per replication) of the sense of responsibility for one’s health on the sense of perceived self-efficacy through dispositional optimism shows that we achieved over 80% power for each model.

## 3. Results

The mean score for generalized self-efficacy among respondents was 28.02 ± 3.67; for dispositional optimism—16.20 ± 3.95, and for health locus of control: internal factors—26.08 ± 3.68, impact of others—21.52 ± 4.06, and random events—19.08 ± 5.36. Significant positive correlations were found between the respondents’ sense of generalized self-efficacy on the one hand, and their dispositional optimism and health locus of control in terms of internal factors on the other, and between dispositional optimism and internal health locus of control. There were also negative correlations between the women’s generalized self-efficacy and their health locus of control in the external factors and random events dimensions, as well as between attribution of health locus of control to external factors and to random events (Table 2).

The mediation analysis presented in Figure 3 revealed significant direct relationships between an internal health locus of control and dispositional optimism (β = 0.14; SE = 0.06; *p* = 0.019) and between an internal health locus of control and self-efficacy (β = 0.35; SE = 0.05; *p* < 0.001). There was also a significant direct impact of the dispositional optimism variable on self-efficacy (β = 0.41; SE = 0.04; *p* < 0.001). The direct impact of internal health control diminished after the inclusion of dispositional optimism as a mediator (β = 0.31; SE = 0.05; *p* < 0.001), which indicates partial mediation. Bootstrap analysis results confirmed the mediating role of dispositional optimism (LLCI = 0.217; ULCI = 0.398).

The mediation analysis presented in Figure 4 demonstrated no association between the impact of others and dispositional optimism (β = 0.06, SE = 0.05, *p* > 0.05). There was an association between the impact of others and self-efficacy (β = –0.14; SE = 0.05; *p* < 0.001). The direct impact of dispositional optimism on self-efficacy was also significant (β = 0.46; SE = 0.04; *p* < 0.001).

The analysis presented in Figure 5 showed an association between the random events dimension and dispositional optimism (β = −0.32, SE = 0.04, *p* < 0.001). There was also an association between random events and self-efficacy (β = −0.08; SE = 0.04; *p* < 0.029). The direct impact of dispositional optimism on self-efficacy was also significant (β = 0.49; SE = 0.05; *p* < 0.001). With the mediator included, the association between random events and self-efficacy became positive, with a value of 0.07 (SE = 0.04, *p* = 0.044; LLCI = 0.020; ULCI = 0.147).

## 4. Discussion

The present findings reveal a number of significant mechanisms observed in women with threatened preterm labor in terms of relationships between factors that constitute their personal resources, i.e., health locus of control, self-efficacy, and dispositional optimism, the latter being a mediator between health locus of control and perceived self-efficacy.

An internal health locus of control (i.e., the belief that one can control one’s own health, as psychophysical condition largely depends on personal behaviors) is positively correlated with self-efficacy. This means that increased internal health control contributes to a stronger sense of self-efficacy, so the pregnancy risk is subjectively perceived as a challenge to be overcome. Women who are thus convinced of their agency with regard to their health are more likely to undertake healthy behaviors, which may contribute to a lower risk of additional complications during the pregnancy. These pregnant women tend to be confident that their behaviors have an impact on their own health as well as that of their baby [30,31]. The perception of various life situations as challenges results from a strong belief in one’s personal capacity to mobilize cognitive, competence, and energy resources for the effective handling of the pregnancy risks that may occur. Increased self-efficacy promotes greater motivation to act, greater involvement in one’s situation, more active ways of coping with stress, and more positive emotions accompanying these activities. An interpretation of the present findings in the context of COR theory indicates that having abundant resources (in the form of internal health control) favors subsequent resource gains in the form of self-efficacy. Dispositional optimism is a mediator in this relationship between an internal health locus of control and self-efficacy. These findings warrant the conclusion that a sense of personal agency in terms of health generates dispositional optimism (i.e., increases confidence or faith in one’s ability to achieve set goals), which in turn strengthens self-efficacy, or the belief in one’s capability to effectively navigate one’s circumstances owing to one’s personal resources (cognitive, competence-related, and/or energy) [3,4,5,6]. Within COR theory, the mediation function of dispositional optimism in the relationship between internal health locus of control and self-efficacy in women with high-risk pregnancy may be interpreted in terms of a gain cycle that is initiated thanks to a high level of personal resources.

A significant mechanism underlies the relationship between the perceived impact of others on one’s health and self-efficacy. This relationship is a negative one, as a stronger conviction of others’ decisive impact on one’s health in a situation of risk to the pregnancy co-occurs with poorer self-efficacy in the women affected (i.e., a stronger sense that they will not cope with their situation due to a low or insufficient level of personal cognitive, competence, and/or energy resources). The perceived limited ability to act in a given situation, on the one hand, considerably reduces the motivation potential for further action in women with high-risk pregnancy, and on the other, it is a significant risk factor for passivity, helplessness, depression, and/or anxiety. In a situation of threatened preterm labor requiring hospitalization, an external health locus of control will significantly reduce motivation with regard to beneficial health habits in the pregnant women and contribute to their poorer psychophysical condition, weaker motivation for undertaking efforts to maintain wellbeing, denial of disease symptoms, as well as poorer adherence to medical recommendations, such as those concerning follow-up, medication-taking, or diet [16,32,33,34,35,36,37,38,39]. The hypothesized negative correlation between an external health locus of control (indicating a low level of personal resources) and self-efficacy has been verified positively, which confirms the mechanism described within COR theory, whereby having scarce resources triggers cycles of further losses, which consequently exacerbate the negative effects of stress. The present findings did not confirm a mediating function of dispositional optimism in the relationship between external health control in women with high-risk pregnancy and their self-efficacy. There was, however, a positive association between dispositional optimism and self-efficacy, which indicates that increases in dispositional optimism (viewing current and future events as favorable rather than unfavorable) generate a stronger sense of self-efficacy (confidence in one’s personal resources that will help one successfully navigate their situation) [18,40,41,42,43,44]. In the context of the mechanism described, it may be suggested that a concurrent increase in dispositional optimism and self-efficacy constitutes a kind of personal resource gain cycle, predisposing women with high-risk pregnancy to treating their current circumstances as a challenge. There is no association, however, between the perceived impact of others on one’s health and the level of dispositional optimism (belief in the possibility of constructive changes occurring in one’s difficult situation). It can be assumed that the fundamental relationship between external health control and the sense of self effectiveness is so strong that the individual’s sense of optimism does not change this relationship. Optimism does not act as a buffer, a resource that would counteract a negative relationship. Pregnant women with an external locus of control have a sense of no influence on the environment, and perceive their own experiences as the result of the actions of factors beyond their control. The level of placing hope in others (e.g., medical staff) is so strong that it does not activate defense mechanisms, buffering in the form of dispositional optimism. Therefore, these women have a reduced sense of self-efficacy. Of course, this relationship must be studied to be able to talk about cause-and-effect relationships.

The present study demonstrated a negative association between the perceived impact of random events on one’s health and self-efficacy. This indicates that the perception of one’s health condition as the result of random events generates a decrease in self-efficacy in women with high-risk pregnancy (i.e., strengthens the conviction that they will not cope with their situation due to insufficient or inadequate personal cognitive, competence, and/or energy resources). In turn, decreases in self-efficacy represent a risk factor for lower motivation for adaptive action, passivity, helplessness, depression, and/or anxiety. The negative correlation found between the perceived impact of random events on one’s health (indicative of a low level of personal resources) and self-efficacy confirms the present hypothesis. At the same time, this finding warrants the reiteration of the COR-based conclusion that a low level of personal resources triggers loss cycles leading to negative outcomes. Dispositional optimism is a mediator in the relationship between the perceived impact of random events on one’s health and perceived self-efficacy. The mediation function of this factor manifests itself in three aspects, namely, mediation by optimism changes the direction of association between the perceived health impact of random events and self-efficacy in women with high-risk pregnancy, so when the level of dispositional optimism is high (i.e., the woman is convinced that current and future events are positive rather than negative), a strong sense that one’s health is influenced by random events coexists with increased self-efficacy. Dispositional optimism (belief in the possibility of transforming a negative situation into a positive one and/or achieving one’s goals) also reduces the perceived impact of random events on one’s health (negative correlation), while simultaneously enhancing self-efficacy (positive correlation), thus leading to increased confidence that one has the personal resources (cognitive, competence, and/or energy) required for effective handling of the situation presented by one’s high-risk pregnancy. Increased self-efficacy promotes greater motivation to act and/or use more active ways of coping with stress, and/or favors more positive emotions accompanying these activities.

## 5. Conclusions

The characteristics of the mediation functions described above warrant the formulation of certain conclusions building on the COR theory. Firstly, dispositional optimism in women with high-risk pregnancy represents a very significant resource for coping with the problems they encounter, due in part to the changed direction of association (from negative to positive) between the perceived impact of external factors (random events) on one’s health and one’s sense of self-efficacy. Secondly, another strength of dispositional optimism lies in the fact that it prevents the cycle of loss caused by the interpretation of random events as having an impact on one’s health. Thirdly, dispositional optimism, as a mediator, triggers a cycle of resource gains (in the form of increased self-efficacy). Pregnant women characterized by optimism maintain a positive future outlook, even when confronted with negative experiences, represented here by threatened preterm labor. Optimists typically interpret negative circumstances as external, temporary, and situational, while positive circumstances are viewed in the opposite way: as personal, lasting, and universal. This is why dispositional optimism is sometimes described as the opposite of hopelessness [3,4,5,6].

In the literature on the subject, many researchers describe pregnancy as a critical, emotionally significant event that necessitates changes in one’s functioning within one’s family and society. High-risk pregnancy, especially one requiring hospitalization, is thus all the more likely to be a source of negative emotions and stress for the woman. Koss et al. describe high-risk pregnancy as a highly stressful situation threatening the life and health of the baby or even that of the pregnant woman [45]. In this situation, the woman is often forced to change her established way of living, plans, and values; and faces such emotions as fear, anxiety, grief, and a sense of danger [1,2].

The present findings add to those from our previous study that analyzed self-efficacy, life orientation, and health locus of control in pregnant women diagnosed with threatened premature labor [46]. One limitation of the present study is the lack of a detailed analysis of personal resources in women hospitalized due to high-risk pregnancy at a point before the complication had occurred and after the completion of pregnancy.

The project has been realized in a correlation scheme, as a consequence of which it is not possible to draw cause and effect conclusions based on the obtained results. The presented results are exploratory. Other limitations of our study result from the research method used and factors that may have influenced the responses provided by these pregnant women, such as their emotional state at the time. Moreover, modifying factors such as support received from loved ones and demographic variables are worth considering in further analyses. Further research should answer the question whether the models presented in the article, are universal in nature, or whether there are mechanisms specific to different populations in the analyzed context.

## Figures and Tables

**Figure 1 ijerph-19-06075-f001:**
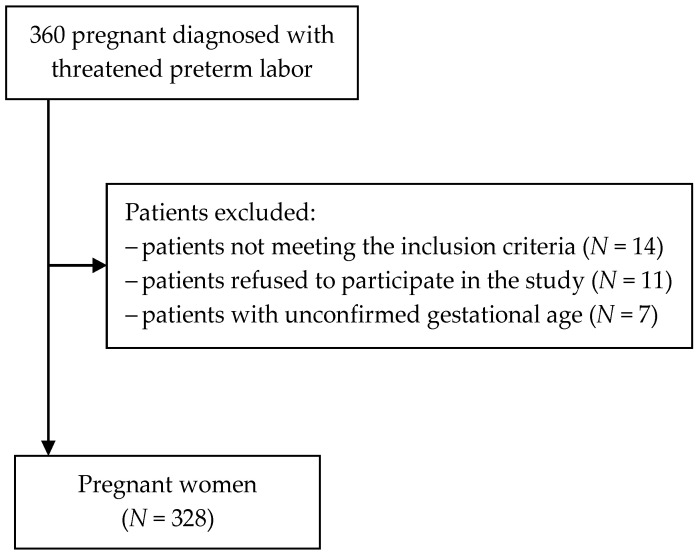
Flowchart of the patient recruitment process.

**Figure 2 ijerph-19-06075-f002:**
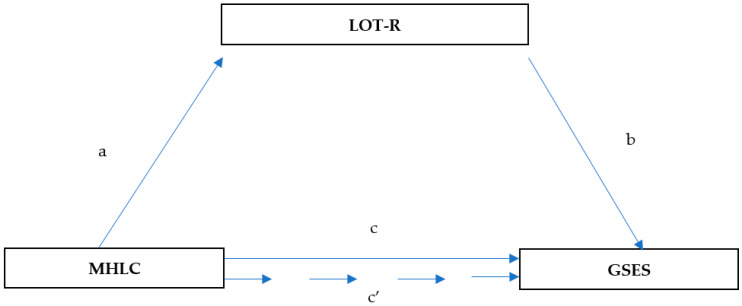
Model of relationships between health locus of control, dispositional optimism, and self-efficacy. (c) The relationship between MHLC and GSES; (a) the relationship between MHLC and LOT-R; (b) the relationship between LOT-R and GSES after controlling for the independent variables; and (c′) the relationship between MHLC and GSES after adding LOT-R as a mediator.

**Figure 3 ijerph-19-06075-f003:**
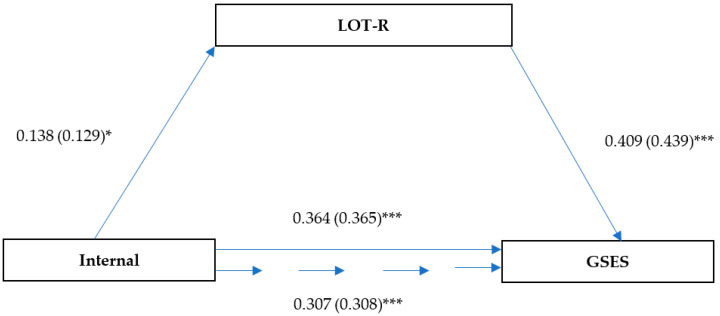
Model of relationships between health locus of control (internal control), dispositional optimism, and self-efficacy. Unstandardized coefficient (standardized coefficient) * *p* < 0.05; *** *p* < 0.001.

**Figure 4 ijerph-19-06075-f004:**
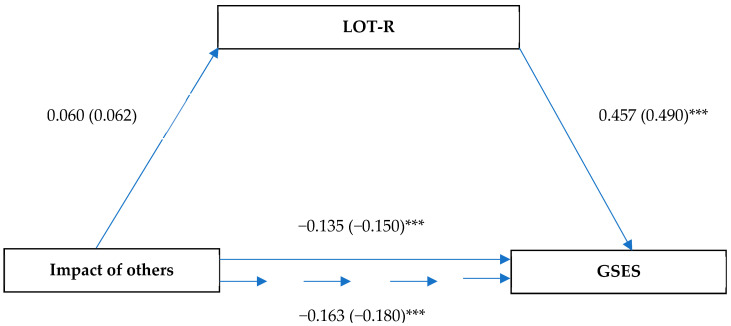
Model of relationships between health locus of control (impact of others), dispositional optimism, and self-efficacy. Unstandardized coefficient (standardized coefficient); *** *p* < 0.001.

**Figure 5 ijerph-19-06075-f005:**
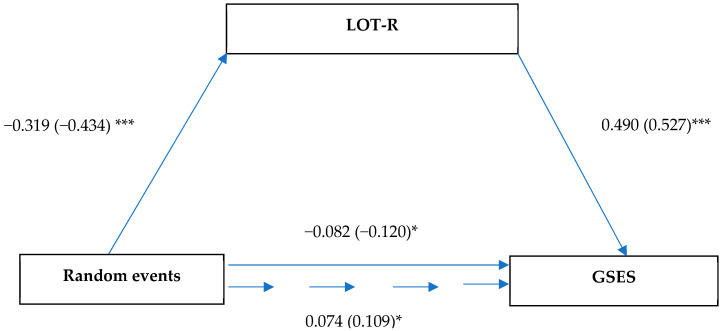
Model of relationships between health locus of control (random events), dispositional optimism, and self-efficacy. Unstandardized coefficient (standardized coefficient); * *p* < 0.05; and *** *p* < 0.001.

**Table 1 ijerph-19-06075-t001:** Characteristics of women in the study.

Characteristics of the Group		Case Group	
		*N*	%
Age	18–25 y/o	95	29.0
	26–35 y/o	190	57.9
	More than 35 y/o	43	13.1
Residence	Urban	179	54.6
	Rural	149	45.4
Relationship status	Married/In a stable relationship	237	72.3
	Single	91	27.7
Education	Other than higher	168	51.2
	Higher	160	48.8
Socioeconomic standing	Satisfying	170	51.8
	Not satisfying	158	48.2
Number of pregnancies	First pregnancy	132	40.2
	Second pregnancy	146	44.5
	Third or subsequent pregnancy	50	15.2
Number of previous deliveries	None	251	76.5
	One	72	22.0
	Two or more	5	1.5
Week of pregnancy	23–27 Hbd	96	29.3
	28–32 Hbd	113	34.5
	32–37 Hbd	119	36.3

**Table 2 ijerph-19-06075-t002:** Descriptive statistics and correlations between the analyzed variables.

		1	2	3	4	5
GSES [1]		-				
LOT-R [2]		0.479 **	-			
MHLC	Internal [3]	0.365 **	0.129 *	-		
	Impact of others [4]	−0.149 **	0.062	0.099	-	
	Random events [5]	−0.120 *	−0.434 **	−0.032	−0.125 *	-
	M	28.02	16.20	26.08	21.52	19.08
	SD	3.67	3.95	3.68	4.06	5.36
	α	0.76	0.74	0.57	0.65	0.69

GSES—generalized self-efficacy scale; LOT-R—Revised Life Orientation Test; MHLC—multidimensional health locus of control scale; α—Alpha Cronbach; M—mean; SD—standard deviation; * *p* < 0.05; and ** *p* < 0.01.

## Data Availability

The datasets generated during and/or analyzed during the current study are available from the corresponding author on reasonable request.

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
