# Peer review of "The Mediating Role of Dispositional Optimism in the Relationship between Health Locus of Control and Self-Efficacy in Pregnant Women at Risk of Preterm Delivery"

_ijerph, 2022, doi:10.3390/ijerph19106075_

Round 1

Reviewer 1 Report

Niewiadomska et al. assess the role of dispositional optimism on health locus of control and self-efficacy in pregnant women with threatened preterm labor. Overall, the study is rigorously executed and clearly communicated in this manuscript. Some minor comments -

How generalizable are the results as respondents were aged 26–35 years (57.9%), urban residents 129 (54.6%), married/in a stable relationship (72.3%), had not completed higher education 130 (51.2%), and had a satisfactory socio-economic standing (51.8%)?

How can authors claim (51.2-57.9)% to be representative of “most” of the respondents?

Could the significant correlations found between the respondents’ sense of generalized self-efficacy and their dispositional optimism and health locus of control in terms of internal factors be due to any other confounding factors?

The authors should explain more clearly why their findings did not confirm a mediating function of dispositional optimism in the relationship between external health control in women with high-risk pregnancy and their self-efficacy.

Author Response

The Authors would like to thank the Reviewer for many valid remarks, comments and suggestions. We have revised our paper accordingly and as requested We have provided below a detailed response to reviewer’ comments.

Point 1: How generalizable are the results as respondents were aged 26–35 years (57.9%), urban residents 129 (54.6%), married/in a stable relationship (72.3%), had not completed higher education 130 (51.2%), and had a satisfactory socio-economic standing (51.8%)? How can authors claim (51.2-57.9)% to be representative of “most” of the respondents?

Response 1: Thank you, for your comment. We have made a correction in the transcript of the results, which now reads: Among the respondents women were the most numerous group, aged 26–35 years (57.9%), urban residents (54.6%), married/in a stable relationship (72.3%), had not completed higher education (51.2%), and had a satisfactory socio-economic standing (51.8%); typically, they were pregnant for the second time (44.5%), nulliparous (76.5%), between weeks 32 and 37 (36.3%) – table 1.

Point 2: Could the significant correlations found between the respondents’ sense of generalized self-efficacy and their dispositional optimism and health locus of control in terms of internal factors be due to any other confounding factors?

Response 2:

We definitely agree with the reviewer that there may be various factors modifying, disturbing the analyzed correlations. These factors may include women's emotional state, their psychophysical condition, the support they receive, etc.  Moreover, we would like to mention that in this study, which was exploratory, we only analyzed the role of optimism as a mediator, without the participation of covariates. However, we thank you for your valuable comment. In the limitations we have pointed out this weakness of our study.

Point 3: The authors should explain more clearly why their findings did not confirm a mediating function of dispositional optimism in the relationship between external health control in women with high-risk pregnancy and their self-efficacy.

Response 3:

We assume that the underlying relationship between external health control and sense of efficacy is so strong that an individual's sense of optimism does not alter this relationship. Optimism does not act as a buffer, a resource to offset the negative relationship. Pregnant women with an external locus of control feel they have no influence on their environment and perceive their own experiences as the result of factors which they can not control. The level of placing hope in others (medical staff) is so strong that it does not activate defense mechanisms, buffering in the form of dispositional optimism. Ultimately, these women therefore have a reduced sense of self-efficacy. Obviously, the relationship needs to be studied in order to talk about cause-and-effect relationships.

The Reviewer's suggestions have been implemented and highlighted in blue. We are grateful for the feedback and the opportunity to respond to the Reviews’ comments and suggestions. We hope the changes we have made improve the overall quality of the paper in line with your expectations.

Reviewer 2 Report

The present study by Niewiadomska ete al. entitled "The Mediating Role of Dispositional Optimism in the Relationship Between Health Locus of Control and Self-Efficacy in Pregnant Women at Risk of Preterm Delivery" describes a Polish cohort study with a total of 328 included pregnant participants. The authors examine the influence of dispositional optimism in women with high-risk pregnancies. They show important socio-psychological effects and significant resources that are influenced by it. They also show that the strength of dispositional optimism lies in the fact that it prevents the cycle of loss caused by the interpretation of random events as having an impact on one person. 
random events as having an impact on one's health. The data shown add to those from their previous study that analysed self-efficacy, life orientation, and health locus of control in pregnant women diagnosed with 
threatened premature labour. The study presented is well presented and clearly written. Other factors, such as the emotional status of the participants, may have had an influence on the results. The study is also a single centre study, so further and larger multi-centre studies are needed to further support the conclusions of the study. 

Author Response

Dear Reviewer,

The Authors would like to thank the Reviewer for valid remarks and suggestions.

Point 1: The data shown add to those from their previous study that analysed self-efficacy, life orientation, and health locus of control in pregnant women diagnosed with threatened premature labour. The study presented is well presented and clearly written. Other factors, such as the emotional status of the participants, may have had an influence on the results. The study is also a single centre study, so further and larger multi-centre studies are needed to further support the conclusions of the study.

Response 1: We agree with the reviewer's opinion that other factors, including emotional state, may influence the results of the study, so we plan to expand our study to include these variables as well as conduct the study multicenter. We appreciate your comment, based on which we have supplemented the limitations section of our study.

The Reviewer's suggestions have been implemented and highlighted in blue. We are grateful for the feedback and the opportunity to respond to the Reviews’ comments and suggestions. We hope the changes we have made improve the overall quality of the paper in line with your expectations.

Reviewer 3 Report

This study examined a mediation effect of dispositional optimism on the association of locus of control with generalized self-efficacy among 328 pregnant women hospitalized due to threatened preterm labor. It was found that dispositional optimism mediated the association. The topic is very suitable for the special issue "Frontiers in Maternal and Reproductive Health". This study reports very precious results collected at clinical settings. However, I have a few serious concerns with the manuscript in its current form. First, the originality of your work is not clear. It would be better to cite previous studies on the subject. Are there any studies on dispositional optimism, locus of control, and/or generalized self-efficacy among pregnant women hospitalized due to threatened preterm labor? Adding references would clarify the originality of your work. Second, it is unclear why a mediation model has been hypothesized. Authors have mentioned COR principles and have proposed the model. I could not understand this flow. More details seem needed.

Author Response

The Authors would like to thank the Reviewer for many valid remarks, comments and suggestions. We have revised our paper accordingly and as requested We have provided below a detailed response to reviewer’ comments.

Point 1: First, the originality of your work is not clear. It would be better to cite previous studies on the subject. Are there any studies on dispositional optimism, locus of control, and/or generalized self-efficacy among pregnant women hospitalized due to threatened preterm labor? Adding references would clarify the originality of your work.

Response 1:

The problem of dispositional optimism, the generalized self-efficacy or the locus of control is ubiquitous in the literature on health / disease. Researchers addressed the relationship between these phenomena in various groups of patients, e.g. with a cancer. To the best of our knowledge, the role of dispositional optimism as a mediator in the relationship between the locus of health control and the sense of self-efficacy among women in complicated  pregnancy has not been analyzed, and our study is the first of its kind. Moreover, we would like to mention that the present findings add to those from our previous study that analyzed self-efficacy, life orientation, and health locus of control in pregnant women diagnosed with threatened premature labor.

Point 2: Second, it is unclear why a mediation model has been hypothesized. Authors have mentioned COR principles and have proposed the model. I could not understand this flow. More details seem needed.

Response 2:

Thank you for your valuable suggestion. In the introduction, we have added a paragraph indicating the relevance of COR theory in the context of the relationships between dispositional optimism, the generalized self-efficacy and locus of health control.

The Reviewer's suggestions have been implemented and highlighted in blue. We are grateful for the feedback and the opportunity to respond to the Reviews’ comments and suggestions. We hope the changes we have made improve the overall quality of the paper in line with your expectations.

Round 2

Reviewer 3 Report

Thank you very much for the opportunity to review the revised manuscript. I feel that the manuscript has been much improved. I think that the revised manuscript could be published, however, I still have one minor comment. 

Minor comment:

I humbly encourage authors to note in the introduction that to the best of their knowledge, the role of dispositional optimism as a mediator in the relationship between the locus of health control and the sense of self-efficacy among women in complicated pregnancy has not been analyzed, and our study is the first of its kind.

If I could not find the description, I am sorry.

Author Response

The Authors would like to thank the Reviewer for suggestions. We strongly believe that the suggested corrections and additions have resulted in an improvement of our manuscript.

We added a sentence in the introduction:

To the best of our knowledge, the role of dispositional optimism as a mediator in the relationship between the locus of health control and the sense of self-efficacy among women in complicated pregnancy has not been analyzed, and our study is the first of its kind.